# TREPH: A Plug-In Topological Layer for Graph Neural Networks

**DOI:** 10.3390/e25020331

**Published:** 2023-02-10

**Authors:** Xue Ye, Fang Sun, Shiming Xiang

**Affiliations:** 1National Laboratory of Pattern Recognition, Institute of Automation, Chinese Academy of Sciences, Beijing 100190, China; 2School of Artificial Intelligence, University of Chinese Academy of Sciences, Beijing 101408, China; 3School of Mathematical Sciences, Capital Normal University, Beijing 100048, China

**Keywords:** graph neural network, graph representation learning, topological data analysis, extended persistent homology

## Abstract

Topological Data Analysis (TDA) is an approach to analyzing the shape of data using techniques from algebraic topology. The staple of TDA is Persistent Homology (PH). Recent years have seen a trend of combining PH and Graph Neural Networks (GNNs) in an end-to-end manner to capture topological features from graph data. Though effective, these methods are limited by the shortcomings of PH: incomplete topological information and irregular output format. Extended Persistent Homology (EPH), as a variant of PH, addresses these problems elegantly. In this paper, we propose a plug-in topological layer for GNNs, termed Topological Representation with Extended Persistent Homology (TREPH). Taking advantage of the uniformity of EPH, a novel aggregation mechanism is designed to collate topological features of different dimensions to the local positions determining their living processes. The proposed layer is provably differentiable and more expressive than PH-based representations, which in turn is strictly stronger than message-passing GNNs in expressive power. Experiments on real-world graph classification tasks demonstrate the competitiveness of TREPH compared with the state-of-the-art approaches.

## 1. Introduction

Graph-structured data are ubiquitous in various domains, including chemistry [1,2], physics [3,4], knowledge graphs [5,6], recommendation [7,8], and social science [9,10]. The analysis of graph data typically involves tasks such as graph classification, node classification, clustering, and link prediction, for which Graph Neural Networks (GNNs) [11] serve as a powerful tool. GNNs are deep learning-based models that take graph data as their input. Most GNNs are constructed by using an iterative message-passing framework [2,10,12,13,14,15,16], which updates a node representation by collecting information from its neighbors. Since messages are exchanged merely in the proximity of each node, these GNN structures are blind to certain global information of the graph. For example, they are unable to detect graph substructures such as cycles (loops) [17]. In fact, it can be shown that the expressive power of message-passing GNNs is bounded above by the one-dimensional Weisfeiler–Leman test [18]. The introduction of Topological Data Analysis (TDA) to the study of GNNs aims to enhance the effectiveness of GNNs by empowering them with global shape-awareness.

TDA is an emerging field that studies the shape of data through the lens of topology [19,20]. The most commonly used tool in TDA is Persistent Homology (PH). Intuitively, the procedure of applying PH can be visualized as scanning an object at multiple scales with respect to a predefined filtration process, which is essentially a process of “growing” the object. The homology functor, which is an algebraic topological construct, is applied to each scale and the inclusions in between, so as to extract the topological information therein. The output of PH typically takes the form of Persistence Diagrams (PDs), which are multisets on the plane. The elements in these PDs encode persistence (i.e., moments of “birth” and “death”) of topological features (connected components, loops, and their analogies in higher dimensions) of the data studied. In the past two decades, TDA has found applications in a growing number of fields, including time series analysis [21], signal processing [22,23,24], image analysis [25,26], physics [27], biology [28,29], chemistry [30,31], material science [32,33,34], finance [35], and so on.

Recently, TDA has been introduced to the area of machine learning [36,37] to enhance the performance of both traditional machine learning methods and deep learning models. In particular, there has been an increased interest in combining TDA with neural networks for graph representation learning. For example, local topological features can be used as side information to rewire the graph and/or reweigh the messages passed between graph nodes during the convolution [38,39]. Although effectiveness is demonstrated well in these methods, the global topological features are still not fully utilized, which could be of great discriminative value. In [40,41], a filtration is built using the geometry of the graph, and the resulting PDs are fed into a neural network for both parameterized vectorization and task-specific prediction. Although the filtrations in these methods capture global topology, they are defined a priori and may not be the optimal choice for the specific task.

Another line of thought is constructing a learnable topological feature extraction layer either serving as a readout operation [42] or being a generic layer in GNNs to endow it with topology awareness [43]. In this manner, the learning signals can backpropagate through the calculation of TDA and adjust the filtrations as needed, enabling the models to be more flexible and adaptable to downstream graph-learning tasks. However, these methods are designed based on PH, which has certain shortcomings innately. One drawback lies in that the output PDs does not enjoy a uniform format, due to the presence of infinite coordinates. As a consequence, necessary compromises, such as tossing away the infinite parts or truncating the data artificially, must be made to enable subsequent use. Another drawback of PH is its inability to fully encode the topological features in graphs, the reason for which will be elaborated in Section 3.1.3.

The aforementioned issues of PH can be solved elegantly by Extended Persistent Homology (EPH) [44]. As a refined version of PH, EPH has an output always containing finite values and is able to extract strictly more topological information than PH. Recent years have seen efforts to integrate EPH with machine learning [38,41,45,46,47]. However, these studies do not incorporate EPH into a framework that is end-to-end trainable.

In this work, we propose a plug-in topological layer, called Topological Representation with Extended Persistent Homology (TREPH). By virtue of EPH, the proposed layer not only retains all the merits held by the aforementioned PH-based layers, but also enjoys strictly stronger expressivity. Furthermore, our designed layer contains a novel aggregation mechanism that takes advantage of the uniformity of EPH for improved efficiency of feature learning. Our contributions in this work are summarized as follows:A plug-in topological layer named TREPH is proposed for GNN, which utilizes EPH for effective extraction of topological features and can be conveniently inserted into any GNN architecture;TREPH is proved to be differentiable and strictly more expressive than PH-based representations, which in turn is strictly stronger than message-passing GNNs in expressive power;By making use of the uniformity of EPH, a novel aggregation mechanism is designed to empower graph nodes with the ability to perform shape detection;Experiments on benchmark datasets for graph classification show the competitiveness of TREPH, achieving state-of-the-art performances.

## 2. Related Work

### 2.1. Graph Neural Networks

The concept of Graph Neural Network [11] is initially outlined in [48] and further illustrated in [49,50]. Early works typically adopt a recurrent neural architecture to learn node representations, assuming that a node keeps exchanging messages with its neighbors until a fixed point is reached. More recently, it has become popular to stack multiple graph convolutional layers to learn high-level node representations [10,12,13,14,15,16]. Most of these models are developed on a message-passing framework [2], in which each node updates its hidden state by collecting messages from its neighbors. Though efficient, a recent study [18] shows that the expressive power of message-passing GNNs is bounded by the Weisfeiler–Leman test (i.e., WL[1]) [51]. In light of this, several works [52,53] design higher-order message-passing schemes to align GNNs with more discriminative WL[*k*] (k≥2) tests.

### 2.2. Topological Data Analysis

Topological data analysis is a field that investigates data by means of algebraic topology. The core concept of TDA is persistent homology, which can be seen as a multiscale representation of topological features. Since TDA is capable of extracting global, structural, and intrinsic properties of data at multiple scales, it is a desirable complement to geometry-based techniques in feature engineering. The synthesis of TDA and machine learning has become a rapidly developing research area [36,37,54]. Most works fall into three categories: vector-based, kernel-based, and neural network-integrated methods.

Vector-based methods focus on the appropriate vectorization of persistence diagrams. The resulting vectors can be used as the input of machine learning algorithms. Typical works involve statistical measurements [55], algebraic combinations [56], persistent entropy [57], Betti curves [58], persistence landscapes [59], and persistence images [60].

Kernel-based methods [61,62,63,64,65,66] aim at designing a kernel function that implicitly embeds PDs into a Reproducing Kernel Hilbert Space (RKHS). Since the kernel measures the inner product of two PDs embedded in the corresponding RKHS, it can be integrated into certain machine learning algorithms. Note that similarity measurements between PDs can also be modified into kernels [67].

More recently, it has been realized that neural networks could benefit from applying TDA to the relevant data. One of the most popular types of data for this application is graph. For instance, one could harness local topological information to rewire the graph and/or reweigh the messages passed between nodes [38,39]. To be precise, for each node, a filtration is defined on a neighborhood using structural information. The similarity between PDs serves as a measurement of the topological similarity between nodes. These methods, however, could fail to utilize the global topology of the graph, which may be critical for downstream tasks. Another approach [40,41,68] is building a filtration with global geometric information extracted from the graph and processing the resulting PDs using a neural network.

The above-mentioned methods use fixed filtrations. A natural direction of improvement is to consider learnable ones instead. A learnable filter function is first proposed in GFL [42] to develop the readout operation. Under their framework, the learning signal can backpropagate through the computation of persistent homology, making it possible to learn a proper filter function for improved performance. Following this line, TOGL [43] proposes a generic topological layer with multiple learnable filter functions. In particular, the PDs computed from these filter functions are transformed and aggregated to graph nodes, producing node-level representations. Such design enables TOGL to be inserted into any GNN architecture for any graph-learning tasks, not limited to graph-level inference. Further, TOGL also proves the superiority of PH over message-passing GNNs in terms of expressivity.

The performance of TDA approaches that use PH, such as those appearing in [39,40,42,43,68], is hindered by two inherent deficiencies (see Section 3.1.3). First, the topological information extracted by PH is incomplete. Second, the infinite coordinates in the PDs turn out to be problematic for certain tasks. For example, in [43], the infinite coordinates were truncated by a chosen upper bound before vectorization. The choice of the upper bound is artificial, twisting the information contained in the PDs. Moreover, each topological feature in PDs is associated only with the vertex whose addition results in its birth, ignoring the death association. This is because some (but not all) features do not die, and thus cannot be handled uniformly with others.

The notion of extended persistent homology arises as a completion of PH [44]. EPH is strictly more expressive than PH (see Theorem 2). The output of EPH enjoys uniformity, being composed solely of points with finite coordinates. Although EPH has found a variety of applications in the study of machine learning [38,41,45,46,47], an end-to-end trainable framework utilizing EPH has yet to be considered.

In this work, we fill this gap by proposing an EPH-enhanced topological layer for GNNs, called TREPH. The performance of TREPH benefits from both the superior expressivity of EPH and an effective aggregation mechanism made possible by the uniformity of EPH.

## 3. Methodology

In this section, we first briefly review the relevant notions of EPH. Then, we describe our TREPH layer in detail. Finally, the differentiability and expressivity of our model are discussed. For a thorough treatment of EPH, we refer interested readers to [19].

Table 1 lists key symbols used in our paper.

### 3.1. Preliminaries

#### 3.1.1. Homology

Homology theory is the study of a class of algebraic invariants called the homology groups, capable of extracting topological information (shape) of an object. We will briefly recall the relevant definitions and properties of homology in what follows, and we refer the readers to [69] for a comprehensive treatment. For our purpose, we consider relative simplicial homology over $Z_{2}=Z/2Z$, where Z stands for the ring of integers, “/” indicates taking quotient, and 2Z={2n|n∈Z}.

Let G=(V,E) be a finite graph with vertex set *V* and edge set *E*. Denote by $C_{0}\left(G\right)$ (resp. $C_{1}\left(G\right)$) the $Z_{2}$-vector space with basis *V* (resp. *E*). Given a subgraph $G^{\prime }=\left(V^{\prime },E^{\prime }\right)$ of *G*, we call $\left(G,G^{\prime }\right)$ a graph pair. Define $C_{0}\left(G^{\prime }\right),C_{1}\left(G^{\prime }\right)$ analogously and set $C_{i}\left(G,G^{\prime }\right)=C_{i}\left(G\right)/C_{i}\left(G^{\prime }\right),i=0,1$. For n∈Z\{0,1}, let $C_{n}\left(G,G^{\prime }\right)=0$. Define the boundary homomorphism $\partial_{1}:C_{1}\left(G\right)\to C_{0}\left(G\right)$ by sending each edge e={v,w} to v+w. Passing to quotients, we obtain a boundary homomorphism $\partial_{1}:C_{1}\left(G,G^{\prime }\right)\to C_{0}\left(G,G^{\prime }\right)$. For n≠1, we let $\partial_{n}:C_{n}\left(G,G^{\prime }\right)\to C_{n-1}\left(G,G^{\prime }\right)$ be zero. The *n*-th (relative) homology group of $\left(G,G^{\prime }\right)$ is then defined as Hn(G,G′)=Ker∂n/Im∂n+1. When $G^{\prime }=\emptyset$, we write $H_{n}\left(G\right)$ for $H_{n}\left(G,G^{\prime }\right)$.

The homology groups encode topological features of the pair $\left(G,G^{\prime }\right)$: the basis of $H_{0}\left(G,G^{\prime }\right)$ is in bijection with the components of *G* not intersecting $G^{\prime }$, while the basis of $H_{1}\left(G,G^{\prime }\right)$ corresponds to the (independent) loops in the quotient graph $G/G^{\prime }$, formed by collapsing the subgraph $G^{\prime }\subseteq G$ to a point.

Given graph pairs $\left(G_{1},G_{1}^{\prime }\right)$ and $\left(G_{2},G_{2}^{\prime }\right)$ such that $G_{1}\subseteq G_{2},G_{1}^{\prime }\subseteq G_{2}^{\prime }$, the inclusion $\left(G_{1},G_{1}^{\prime }\right)↪\left(G_{2},G_{2}^{\prime }\right)$ induces a canonical homomorphism $H_{n}\left(G_{1},G_{1}^{\prime }\right)\to H_{n}\left(G_{2},G_{2}^{\prime }\right)$ for each *n*.

The homology groups encode topological features of *G*: the basis of $H_{0}\left(G\right)$ is in bijection with the components of *G*, while the basis of $H_{1}\left(G\right)$ corresponds to (independent) loops. For this reason, we refer to components and loops of a graph as its 0-dimensional and 1-dimensional topological features, respectively.

#### 3.1.2. Persistent Homology

The discrete nature of homology groups hinders their application in data analysis. One way to introduce continuity is to consider a filter function *f*. Let G=(V,E) be a graph with vertex set *V* and edge set *E*, and f:V→R be a function on *V*. For each a∈R∪{∞}, define the sublevel subgraph $G_{a}=\left(V_{a},E_{a}\right)$, where $V_{a}=f^{-1}\left(-\infty ,a\right]$ and $E_{a}$ denotes the set of edges in *E* joining vertices in $V_{a}$. As *a* increases, this produces the sublevel filtration, which is a family of nested graphs indexed by R (see Figure 1a for an illustration). Regarding a∈R as a temporal parameter and the filtration as a process of expanding sublevel subgraphs, the theory of persistent homology reads the shape of this process using the homology groups of $G_{a}$’s.

The output of PH is specified by pairs of the form (v,e), where v∈V and e∈E. Each pair is interpreted as a component born when *v* is added during the filtration and dies (merges into an older component) with the addition of *e*. Such a feature (component) is called non-essential. Each unpaired vertex corresponds to a component born with the addition of this vertex and never dies. Similarly, an unpaired edge corresponds to a loop born when the edge is added and lives forever. We call a feature that never dies essential.

To utilize the output of PH in data analysis, one usually resorts to a representation called persistence diagrams (PD), which are multisets on the extended plane R×(R∪{∞}). The 0-dimensional PD consists of two types of elements:For each pair (v,e) such that *v* (resp. *e*) is added at time *a* (resp. *b*), we place a point on the plane with coordinate (a,b);For each unpaired vertex *v* added at time *a*, we place a point (a,∞).

The 1-dimensional PD is constructed by placing for each *e* added at time *a* a point (a,∞). An example is given in Figure 1c.

#### 3.1.3. Extended Persistent Homology

There are two drawbacks of PH. First, the output data do not enjoy a uniform format. The essential topological features are specified by one coordinate, while non-essential features are determined by two. One could mitigate this by either treating them separately [42] or truncating the second coordinate using an artificial finite upper bound [43,68]. However, there does not exist a natural way to handle both types within a uniform framework.

The second issue concerns completeness. The birth time of an essential feature corresponds to either the minimum of *f* in a component for $H_{0}$ or the maximum of *f* in a loop for $H_{1}$. By symmetry, we should also consider the maximum in a component and the minimum in a loop. While the PDs of −f contain this information, there is no indication of how the maxima and minima are matched. We shall see in the following how EPH remedies this problem.

The above drawbacks can be overcome with extended persistent homology. Denote by $\bar{R}$ the set of real numbers with reversed order. For b∈R, let $\bar{b}$ be the corresponding element in $\bar{R}$. Given a filter function *f* on G=(V,E), for each $\bar{a}\in \bar{R}$, we define the superlevel subgraph $G^{\bar{a}}=\left(V^{\bar{a}},E^{\bar{a}}\right)$, where $V^{\bar{a}}=f^{-1}\left[a,\infty \right)$ and $E^{\bar{a}}$ denotes the set of edges joining vertices in $V^{\bar{a}}$. These subgraphs form the superlevel filtration which expands in the direction of $\bar{R}$. Essentially, EPH scans the graph bottom-up using the sublevel filtration and then top-down using the superlevel filtration (Figure 1a) and studies the topological features of this process.

The output of EPH is also a pairing [19], which we explain below. Denote $\bar{V},\bar{E}$ as copies of V,E. For v∈V,e∈E, let $\bar{v},\bar{e}$ be the corresponding element in $\bar{V}$ and $\bar{E}$. The motivation behind this is to treat *v* and $\bar{v}$ as the same vertex scanned in the sublevel filtration and superlevel filtration, respectively. The same works for *e* and $\bar{e}$. Elements of $V\cup E\cup \bar{V}\cup \bar{E}$ are called algebraic simplices, since they serve as the basis in the matrix reduction algorithm for computing EPH. Each algebraic simplex *x* is assigned a value $f\left(x\right)\in R\cup \bar{R}$, indicating its time of entrance into the corresponding filtration, in other words, sublevel filtration if x∈V∪E and superlevel filtration if $x\in \bar{V}\cup \bar{E}$.

The output of EPH is a bijection (pairing) between $V\cup \bar{E}$ and $E\cup \bar{V}$. A pair is of the form $\left(v,e\right),\left(v_{1},\bar{v_{2}}\right),\left(e_{1},\bar{e_{2}}\right)$ or $\left(\bar{v},\bar{e}\right)$. Each pair (α,β) corresponds to a topological feature of a certain dimension persisting from f(α) to f(β). For each of the four types of pairs, the collection of (f(α),f(β))’s forms a multiset on the respective plane ($R\times R,R\times \bar{R}$ or $\bar{R}\times \bar{R}$). The four multisets are called persistence diagrams and are denoted by ${\mathrm{Ord}}_{0},{\mathrm{Ext}}_{0},{\mathrm{Ext}}_{1}$ and ${\mathrm{Rel}}_{1}$, respectively (Figure 1b). We provide a topological interpretation of each:A pair (v,e) is associated with a component (0-dimensional). The points (f(v),f(e))∈R×R form the diagram ${\mathrm{Ord}}_{0}$;A pair $\left(v_{1},\bar{v_{2}}\right)$ is of dimension 0. Geometrically, this means $v_{1}$ and $v_{2}$ are where *f* obtains minimum and maximum in a component of *G*. The points $\left(f\left(v_{1}\right),f\left(\bar{v_{2}}\right)\right)\in R\times \bar{R}$ form the diagram ${\mathrm{Ext}}_{0}$;A pair $\left(e_{1},\bar{e_{2}}\right)$ is of dimension 1. In this case, there is a loop whose maximum and minimum are obtained at $e_{1}$ and $e_{2}$, respectively. The points $\left(f\left(e_{1}\right),f\left(\bar{e_{2}}\right)\right)\in R\times \bar{R}$ form the diagram ${\mathrm{Ext}}_{1}$;A pair $\left(\bar{v},\bar{e}\right)$ is of dimension 1. It is not easy to illustrate the geometric meaning of such a pair directly. Fortunately, it can be interpreted as a 0-dimensional feature of −f that persists from −f(v) to −f(e) by the symmetry of extended persistence (see [19] p.162). The points $\left(f\left(\bar{v}\right),f\left(\bar{e}\right)\right)\in \bar{R}\times \bar{R}$ form the diagram ${\mathrm{Rel}}_{1}$.

The output of EPH is uniform: all algebraic simplices are paired and each point (feature) has two finite coordinates. EPH completes PH in the following sense. For a pair (G,f), PH outputs two persistence diagrams of dimension 0 and 1, respectively. The 0-dimensional PD consists of ${\mathrm{Ord}}_{0}$ and a point (a,∞) for each $\left(a,\bar{b}\right)$ in ${\mathrm{Ext}}_{0}$. The 1-dimensional PD consists of a point (a,∞) for each $\left(a,\bar{b}\right)$ in ${\mathrm{Ext}}_{1}$. By taking the PDs of (G,±f) with respect to PH and using the symmetry of EPH, one could recover ${\mathrm{Ord}}_{0},{\mathrm{Rel}}_{1}$ and individual numbers in the coordinates of ${\mathrm{Ext}}_{0},{\mathrm{Ext}}_{1}$. Yet the pairing of those numbers is lost. Thus, EPH extracts strictly more information than PH. A concrete example demonstrating the distinguishing power of EPH over that of PH is presented in the proof of Theorem 2.

### 3.2. Treph

The architecture of TREPH is shown in Figure 2. A graph G=(V,E) with node features $X\in R^{N\times d}$ is fed as its input. Through a Filtration module F, $d_{f}$ filter functions are learned. By applying EPH to each of the filter functions, we obtain the pairings and diagrams from different scanning perspectives. Then, a Vectorization module V is responsible for transforming each point in the diagrams into a $d_{v}$-dimensional vector. These vectors, together with the pairings, are then fed into a novel Aggregation module A to gather the vectorized topological features to their corresponding nodes on the graph. After the Aggregation, a residual connection and a fully connected (FC) layer are used to obtain the final output $X^{\prime }\in R^{N\times d^{\prime }}$.

The Filtration module F is designed as a GIN-ϵ [18] graph convolutional layer followed by a two-layer MLP (multilayer perceptron). Skip connection by concatenation is used after the convolution. The sigmoid function $\sigma \left(x\right)=1/\left(1+e^{-x}\right)$ is adopted as the last activation, thus mapping values into range [0,1] for efficient learning in Vectorization.

For each filter function generated by F, EPH can be calculated by using the matrix reduction algorithm described in [19]. Denote by eph the process of mapping all the filter functions to their diagrams and pairings. The output consists of $4d_{f}$ PDs $\left\{D^{\left(k\right)}|1\leq k\leq d_{f},D\in \left\{{\mathrm{Ord}}_{0},{\mathrm{Rel}}_{1},{\mathrm{Ext}}_{0},{\mathrm{Ext}}_{1}\right\}\right\}$, and their corresponding pairings $\left\{\left(\alpha_{p}^{b},\alpha_{p}^{d}\right)|p\in \cup_{D,k}D^{\left(k\right)}\right\}$.

To make better use of the points in the diagrams, a Vectorization module V is designed to map each point $p\in \cup_{D,k}D^{\left(k\right)}$ to a high-dimensional vector representation $x_{p}\in R^{d_{v}}$. More concretely, for each type of diagram, a total of $d_{v}$ coordinate functions are learned. In this work, we choose the rational hat structure element [40] as our coordinate function:(1)$$s\left(p\right)=\frac{1}{{1+∥p-c∥}_{1}}-\frac{1}{{1+||r|-∥p-c∥}_{1}|},$$
where $p\in R^{2}$ is a point in the diagrams, and $c\in R^{2},r\in R$ are the learnable parameters.

The main idea behind our Aggregation module A is associating each point *p* in the diagrams to the geometric locations (nodes) marking its birth and death. The vectorized topological feature $x_{p}$ is then aggregated to those locations correspondingly. Given the pairing $\left(\alpha_{p}^{b},\alpha_{p}^{d}\right)$ of *p*, the problem translates into defining a locating function $v:V\cup \bar{V}\cup E\cup \bar{E}\to V$. To this end, the sublevel filtration can be regarded as a process of incremental addition of nodes, one at a time, and edges joining existing nodes and the new one. Thus, each element x∈V∪E can be associated with a node v(x) marking its location of entrance. Similarly, each $x\in \bar{V}\cup \bar{E}$ can be assigned an $\bar{v}$ marking its entrance into the superlevel filtration, and we define v(x)=v in this case. Hence, we construct A by aggregating each $x_{p}$ to $v(\alpha_{p}^{b})$ and $v(\alpha_{p}^{d})$, thus empowering graph nodes with the awareness of certain “shapes” in the graph. The details are shown in Algorithm 1, where we also employ the batch normalization [70] and the ReLU activation (ReLU(x)=max(x,0)) for efficient learning.
**Algorithm 1:** Aggregation.
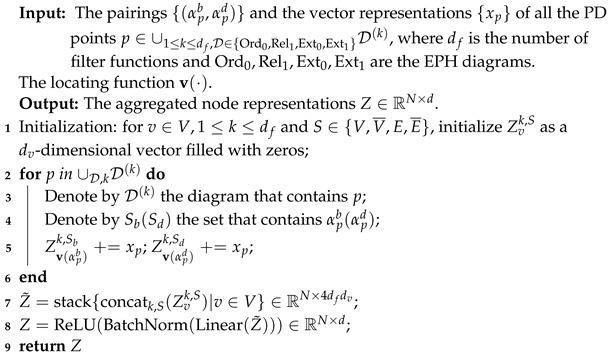


### 3.3. Differentiability and Expressive Power

The following theorem ensures the differentiability of the TREPH layer. Thus, TREPH is end-to-end-trainable with gradient-based optimization.

**Theorem 1**.*If the filter functions involved depend differentiably on a set of parameters* Θ, *then the output of the TREPH layer is differentiable with respect to* Θ *wherever the filter functions are injective (one-to-one).*

**Proof.** Suppose the filter functions F(G,X)=F(G,X,Θ) depend differentiably on a set of parameters Θ. For $\Theta_{0}$ such that $F(G,X,\Theta_{0})$ consists of injective filter functions, we would like the composition A∘V∘eph(G,F(G,X,Θ)) to be differentiable with respect to Θ at $\Theta_{0}$.It is not hard to see that F(G,X,·) and A∘V are differentiable (the latter with respect to the coordinates of points on the PDs). The desired result for PH in place of EPH is proved in [42] (Lemma 1).The key idea for their proof is the following observation. Given $\Theta_{0}$ such that $F(G,X,\Theta_{0})$ consists of injective functions, for Θ in a small enough neighborhood of $\Theta_{0}$, the order of vertices with respect to the filter functions is invariant, so are the pairing (between vertices and edges) and the vertices associated to each pair (i.e., marking the entrance of that vertex/edge). Since each pair determines a point on the PD whose coordinates are values of filter functions on the vertex associated with the pair, differentiability follows. The above argument remains valid for EPH, with pairing now between algebraic simplices. The rest of the proof for [42] Lemma 1 carries over. □

It is shown that PH-enhanced GNNs have strictly stronger expressivity than standard GNNs in terms of the Weisfeiler–Leman test (WL[1]) ([43] Theorem 2). The same can be said for EPH since EPH extracts strictly more information than PH.

To compare the expressivity of PH and EPH, we consider the task of distinguishing pairs of the form (G,f) where *f* is a real-valued function defined on the node set of *G*. By allowing for arbitrary functions (instead of functions generated by the WL algorithm [51]), we take into account the actual input of PH/EPH layers in GNNs.

**Theorem 2**.
*EPH is strictly more expressive than PH. That is, there exist pairs $\left(G_{1},f_{1}\right),\left(G_{2},f_{2}\right)$ such that both $\left(G_{1},f_{1}\right),\left(G_{2},f_{2}\right)$ and $\left(G_{1},-f_{1}\right),\left(G_{2},-f_{2}\right)$ have identical PDs with respect to PH. Yet the PDs of $\left(G_{1},f_{1}\right)$ and $\left(G_{2},f_{2}\right)$ with respect to EPH differ.*


**Proof.** Consider two pairs $\left(G_{1},f_{1}\right)$ and $\left(G_{2},f_{2}\right)$ illustrated in Figure 3a. The PDs of $\left(G_{1},f_{1}\right)$ and $\left(G_{2},f_{2}\right)$ with respect to PH coincide (Figure 3b). The PDs of $\left(G_{1},-f_{1}\right)$ and $\left(G_{2},-f_{2}\right)$ with respect to PH are also identical (Figure 3c). Yet $\left(G_{1},f_{1}\right)$ and $\left(G_{2},f_{2}\right)$ have distinct ${\mathrm{Ext}}_{0}$ and ${\mathrm{Ext}}_{1}$, as illustrated in Figure 3d,e. Here is a brief explanation of what transpires. For both $\left(G_{1},f_{1}\right)$ and $\left(G_{2},f_{2}\right)$, the minima of the two components are 1 and 2, while the maxima are 4 and 5. Thus, the infinite part of the 0-dimensional PDs cannot distinguish them. However, for $\left(G_{1},f_{1}\right)$, the maximum 5 is matched with the minimum 1, and 4 is matched with 2. For $\left(G_{2},f_{2}\right)$, the matchings are (5,2) and (4,1). This matching is captured by ${\mathrm{Ext}}_{0}$, resulting in different PDs. The case for ${\mathrm{Ext}}_{1}$ can be explained analogously. □

## 4. Experiments

### 4.1. Datasets

Our model is evaluated on six graph classification benchmark datasets: REDDIT-BINARY, IMDB-BINARY, IMDB-MULTI, COX2, DHFR, and NCI1 [71]. The first three datasets are derived from social networks, while the rest are molecules collected from medical or biological domains. A statistical summary of these datasets is shown in Table 2.

Ten-fold cross-validation splitting is performed on each dataset. In each fold, 10% of the training set is reserved for validation. For fair comparisons with baselines, experiments in Section 4.2.1 and Section 4.3 are conducted on ten folds of the datasets, and we report the mean and standard deviation of the testing accuracies across these folds. In Section 4.2.2, Section 4.4 and Section 4.5, where we focus on factors that affect the performance of our layer, a fixed fold of each dataset is used for experiments under the consideration of limited resources.

### 4.2. Structure-Based Experiments

The experiments in this section are performed in a purely structure-based context, i.e., the input node features are set as uninformative (vectors of all ones) for all datasets. The main purpose of such design is to examine the practical capacity of TREPH in structural capturing compared with classical message-passing GNNs. To be specific, we first investigate the effects of inserting TREPH into GCNs [10], GATs [16], and GINs [18] and then study the influences of different positions where TREPH is placed.

#### 4.2.1. TREPH in GNNs

Three groups of comparative experiments are conducted on six datasets. In each group, a four-layer GNN of a certain type (4-GCN, 4-GAT, or 4-GIN) is used as the baseline, and the other model is designed by substituting TREPH for the first layer of the baseline (3-GCN-1-TREPH, 3-GAT-1-TREPH, or 3-GIN-1-TREPH). Table 3 shows that the models equipped with TREPH generally perform better than the baselines of the same group (TREPH shows superiority in 14 out of 18 comparisons). Such a phenomenon indicates the relative advantages of TREPH over message-passing GNNs in terms of structural learning. Note that in GAT cases, where we experience severe gradient-vanishing problems on some of the datasets, by applying TREPH for the first layer, these training issues can be fixed to some extent. The number of parameters for each model on each dataset is listed in Table 4. Similar to the GNN baselines, the number of parameters of TREPH-equipped models is independent of the graph sizes. A little increase of the parameters in IMDB-MULTI comes from the classifier, which is caused by the increase of classes to predict.

#### 4.2.2. Study of TREPH Positions

TREPH is a plug-in layer, which means that it can be inserted into any position of any GNN architecture. To investigate the influences of placing TREPH at different positions, three groups of experiments are implemented. In each group, we consider substituting TREPH for the first, second, third, or fourth layer in certain four-layer GNNs. The results are displayed in Table 5. It can be seen that the best position to place TREPH differs from case to case. For GINs, the second layer seems to be a good choice for TREPH in general. However, such placement is not appropriate for GATs due to the training issues mentioned in Section 4.2.1. In these cases, the first layer is a more reliable place for TREPH to avoid degradation of the model. For GCNs, there seems to be no uniform standard on which position is preferable. The best position depends on the dataset being processed.

### 4.3. Comparison with State-of-the-Art Methods

There are nine state-of-the-art baseline methods considered in our work for comparison. WL and WL-OA [72] are two kernel methods inspired by the Weisfeiler–Leman test. PATCHY-SAN [73], DGCNN [74], and SAT [75] are three end-to-end-trainable neural networks that operate on graphs. SV [66], PersLay [41], GFL [42], and TOGL [43] are four TDA-based approaches. Note that all TDA-based approaches (including ours) did not utilize the node labels/attributes possibly provided by the datasets, while other methods did. To align with previous TDA-based methods, we employ a learnable embedding layer to generate input from node degrees for all datasets except for REDDIT-BINARY, whose input is set as uninformative.

Table 6 shows the graph classification results of each method on six benchmark datasets. It can be seen that our model performs comparably with state-of-the-art methods, and in particular, it outperforms all TDA-based approaches. On social networks, which do not have node labels/attributes, our model exhibits superiority over other methods. On molecular datasets, for which node labels/attributes are available, our model still outperforms others on COX2 and DHFR, despite the fact that we do not utilize that information during training and inference. The only exception is NCI1. We conjecture that the node labels of NCI1 might contain critical information for effective prediction, which is somewhat complementary to the structural information. Nevertheless, our method has a clear advantage over those TDA-based approaches, proving the power of TREPH in extracting discriminative topological features.

### 4.4. Ablation Study

Two variant models are designed to study the effect of replacing EPH with PH. In fact, we go a step further and compare the EPH of (G,f) to the PH of (G,f) and (G,−f) combined. As remarked near the end of Section 3.1.3, this replacement amounts to the following two operations:Each point (a,b) of ${\mathrm{Ext}}_{0}$ is broken into a point (a,∞) for the 0-dimensional PD of *f* and a point (−b,∞) for the 0-dimensional PD of −f;Each point (c,d) of ${\mathrm{Ext}}_{1}$ is broken into a point (c,∞) for the 1-dimensional PD of *f* and a point (−d,∞) for the 1-dimensional PD of −f.

Note that the information contained in ${\mathrm{Ord}}_{0}$ and ${\mathrm{Rel}}_{1}$ is preserved perfectly by the replacement.

The first variant model TREPH-noext treats infinite points on the PDs separately from finite ones. To be precise, it regards a point (on one of the PDs) of the form (a,∞) as a∈R, vectorizes accordingly, and assigns the vector to the algebraic simplex where the topological feature corresponding to the point is born. The finite portion of the 0-dimensional PDs of *f* and −f are identical to ${\mathrm{Ord}}_{0}$ and ${\mathrm{Rel}}_{1}$ of *f*, respectively (modulo symmetry for ${\mathrm{Rel}}_{1}$), and are treated in the same way as TREPH. Note that there are no finite points in the 1-dimensional PDs.

The second variant model TREPH-noext-trunc truncates infinite coordinates by replacing each point (a,∞) on the PDs of *f* (respectively, −f) by (a,maxf) (respectively, (a,max−f)). Each resulting PD consists solely of finite planar points, which can be vectorized uniformly. Each vector obtained is aggregated to the algebraic simplex corresponding to the birth of that feature, ignoring the simplex of death. This is because truncated points have no simplex of death.

Experimental results comparing the performance of TREPH with that of its variants on six datasets are displayed in Figure 4. It can be seen that TREPH compares favorably with both of its variants on all six datasets. This demonstrates the superiority of the expressivity of EPH.

### 4.5. Analysis of Hyperparameters

Two types of hyperparameters are analyzed. One is the number of filter functions $d_{f}$, the other is the dimension of vectorized representations $d_{v}$. Figure 5 shows the experimental results on six datasets under different choices of hyperparameters. On at least four datasets (REDDIT-BINARY, COX2, DHFR, and NCI1) out of six, there is an evident trend that larger $d_{f}$’s tend to provide better performances, indicating that increasing the number of topological features extracted could enhance the capability of our model. Similarly, experiments on the same four datasets show that increasing $d_{v}$ results in better performances. The remaining two datasets (IMDB-B, IMDB-M) are significantly denser (having a larger edge-to-node ratio) than the others. We speculate that this results in their vulnerability to overfitting. The speculation is validated by the experiments on these datasets, as the increase of two independent hyperparameters leads to poorer performance.

### 4.6. Implementation Details

The input node dimension, the number of filter functions $d_{f}$, and the length of vector representation $d_{v}$ are set as 64, 8, and 32, respectively. The node representations are average-pooled before being passed to a two-layer MLP with hidden dimension 32 for final prediction. Cross-entropy loss is used for training. The experiments are implemented in PyTorch [76] and optimized with the Adam algorithm [77]. The learning rate is set as 0.001 initially and halved if the validation accuracy does not improve after 10 epochs. We stop training when the learning rate is less than $10^{-5}$ or the number of epochs reaches 200.

### 4.7. Complexity Analysis

The original matrix reduction algorithm for computing EPH [19] runs in $O(n^{3})$ with n=|V∪E| in the worst case. In our experiments, a CUDA (https://developer.nvidia.com/cuda-toolkit accessed on 10 January 2023) programmed variant is designed for speedup. Timing experiments are conducted on the dataset REDDIT-BINARY (see Figure 6), which shows that our algorithm scales well in practice.

We remark that the computational efficiency of the TREPH layer could be further improved by optimizations of EPH computation. For example, one could resort to neural algorithm execution. In fact, a GNN framework for approximating the PDs of EPH is proposed in [78]. Their method, however, cannot be directly applied to our model, since the pairings of EPH cannot be obtained from their algorithm. A neural approximation that outputs the pairings of EPH is left as a subject for future research.

## 5. Conclusions

In this paper, we propose TREPH, which is a plug-in topological layer for GNN. By virtue of the uniformity of EPH, a novel aggregation mechanism is designed to collate topological features to their corresponding nodes on the graph, thus empowering graph nodes with the ability of shape detection. We have proved that the proposed layer is differentiable and strictly more expressive than PH-based representations, which in turn is stronger than message-passing GNNs in expressivity. Experiments on real-world graph classification tasks demonstrate the effectiveness and competitiveness of our model compared with the state-of-the-art methods.

For future research, we could consider speeding up the computation of our layer by developing a neural approximation algorithm for computing the pairings of EPH. In addition, we plan to explore the application of our proposed layer in graph learning tasks beyond graph classification, such as node classification and link prediction.

## Figures and Tables

**Figure 1 entropy-25-00331-f001:**
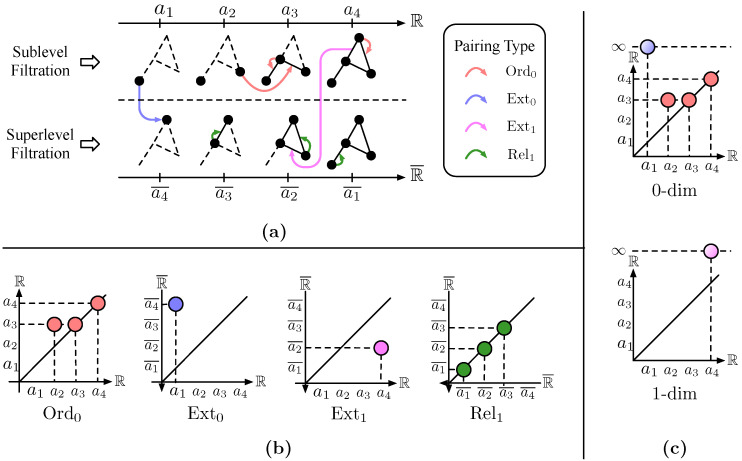
(**a**) Illustration of the sublevel and superlevel filtrations, as well as the pairings of extended persistent homology. (**b**,**c**) The persistence diagrams with respect to extended persistent homology and persistent homology, respectively.

**Figure 2 entropy-25-00331-f002:**
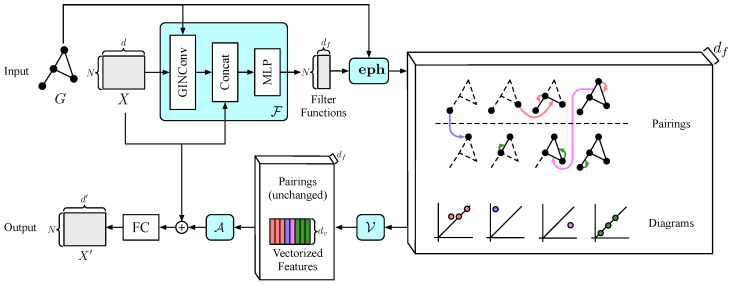
The architecture of TREPH.

**Figure 3 entropy-25-00331-f003:**
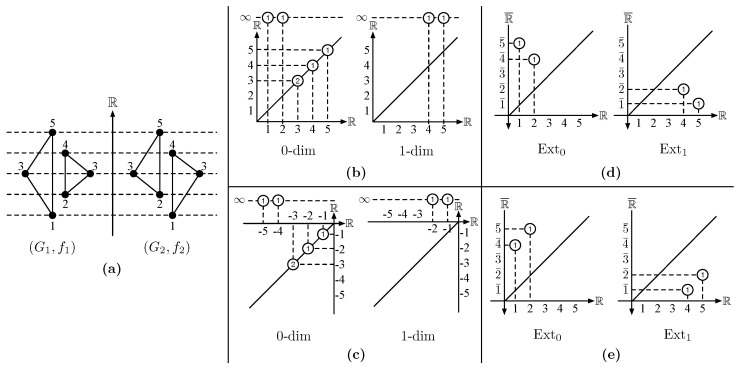
(**a**) Two graphs $G_{1}$ and $G_{2}$ are presented side by side, equipped with filter functions $f_{1}$ and $f_{2}$, respectively. (**b**) PDs of both $\left(G_{1},f_{1}\right)$ and $\left(G_{2},f_{2}\right)$ with respect to PH. The numbers in circles represent multiplicity. (**c**) PDs of both $\left(G_{1},-f_{1}\right)$ and $\left(G_{2},-f_{2}\right)$ with respect to PH. (**d**) ${\mathrm{Ext}}_{0}$, ${\mathrm{Ext}}_{1}$ of $\left(G_{1},f_{1}\right)$. (**e**) ${\mathrm{Ext}}_{0}$, ${\mathrm{Ext}}_{1}$ of $\left(G_{2},f_{2}\right)$.

**Figure 4 entropy-25-00331-f004:**
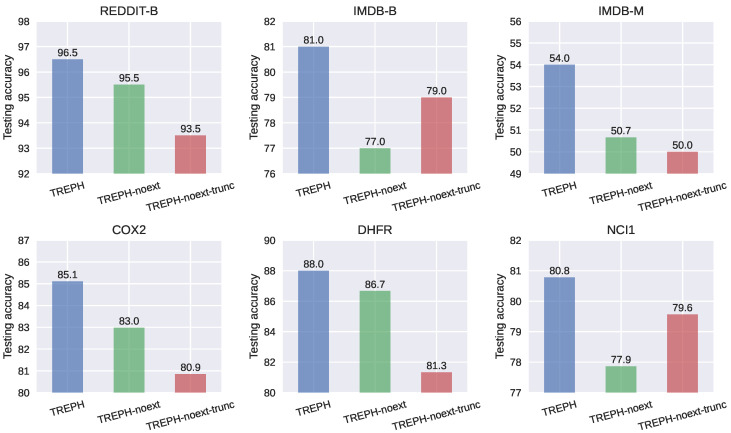
Ablation study of TREPH on six datasets.

**Figure 5 entropy-25-00331-f005:**
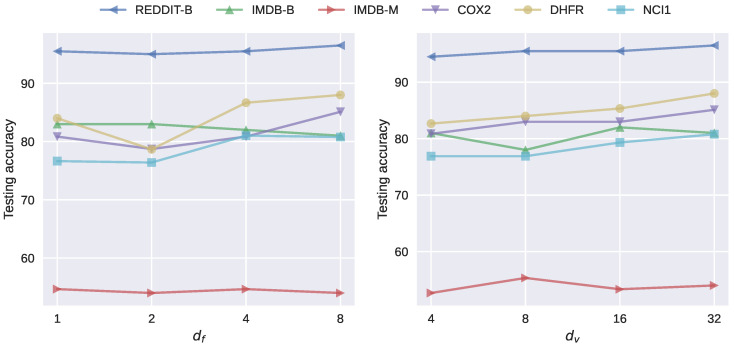
Analysis of the hyperparameters $d_{f}$ and $d_{v}$ in TREPH.

**Figure 6 entropy-25-00331-f006:**
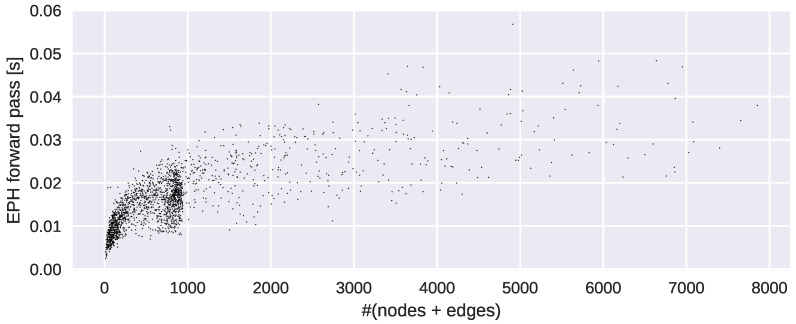
Timing experiments on REDDIT-BINARY.

**Table 1 entropy-25-00331-t001:** A summary of key symbols used in this paper.

Symbol	Meaning
*G*	An undirected graph.
*V*	The vertex set of *G*.
*E*	The edge set of *G*.
*N*	The cardinality of *V*.
$X\in R^{N\times d}$	The input node features of *G*.
Z	The ring of integers.
Ker	The kernel of a homomorphism.
Im	The image of a homomorphism.
*f*	A function V→R.
$G_{a}=\left(V_{a},E_{a}\right)$	The sublevel subgraph of *G* at *a*.
$\bar{R}$	The reversed real line.
$\bar{a}$	An element a∈R regarded as an element of $\bar{R}$.
$\bar{V},\bar{E}$	Copies of V,E.
$\bar{v},\bar{e}$	Elements of $\bar{V},\bar{E}$.
$G^{\bar{a}}=\left(V^{\bar{a}},E^{\bar{a}}\right)$	The superlevel subgraph of *G* at $\bar{a}$.
${\mathrm{Ord}}_{0},{\mathrm{Ext}}_{0},{\mathrm{Ext}}_{1},{\mathrm{Rel}}_{1}$	The four Persistence Diagrams for EPH.
F	The Filtration module.
V	The Vectorization module.
A	The Aggregation module.
eph	The process of computing EPH for all the filter functions.
$d_{f}$	The number of filter functions.
$d_{v}$	The dimension of vectorized representations of PD points.

**Table 2 entropy-25-00331-t002:** A summary of the datasets used in this paper.

Datasets	REDDIT-B	IMDB-B	IMDB-M	COX2	DHFR	NCI1
Graphs	2000	1000	1500	467	467	4110
Classes	2	2	3	2	2	2
Avg. #Nodes	429.63	19.77	13.00	41.22	42.43	29.87
Avg. #Edges	497.75	96.53	65.94	43.45	44.54	32.30

**Table 3 entropy-25-00331-t003:** Graph classification results in the study of inserting TREPH into GNNs under a purely structure-based setting. The best results within each group are emphasized in **bold** font.

Method	REDDIT-B	IMDB-B	IMDB-M	COX2	DHFR	NCI1
4-GCN	92.9±1.9	65.1±5.1	39.9±3.5	77.1±8.5	75.7±4.3	74.7±1.8
3-GCN-1-TREPH	92.6±1.4	71.6±3.9	48.1±2.6	79.2±2.6	77.9±2.6	76.7±1.6
4-GAT	50.0±0.0	50.0±0.0	31.7±4.4	72.4±17.1	61.0±0.5	50.0±0.1
3-GAT-1-TREPH	90.1±3.8	70.9±4.4	48.3±2.8	75.8±7.3	77.9±4.3	76.5±1.6
4-GIN	91.2±2.4	71.1±3.7	45.3±4.2	82.7±2.2	80.7±2.6	77.2±1.3
3-GIN-1-TREPH	90.1±1.1	72.0±3.8	47.4±3.4	80.1±2.9	79.6±4.1	77.6±1.6

**Table 4 entropy-25-00331-t004:** Number of parameters for GNN baselines and TREPH-equipped models on six datasets.

Method	REDDIT-B	IMDB-B	IMDB-M	COX2	DHFR	NCI1
4-GCN	19,298	19,298	19,331	19,298	19,298	19,298
3-GCN-1-TREPH	102,891	102,891	102,924	102,891	102,891	102,891
4-GAT	19,810	19,810	19,843	19,810	19,810	19,810
3-GAT-1-TREPH	103,275	103,275	103,308	103,275	103,275	103,275
4-GIN	36,454	36,454	36,487	36,454	36,454	36,454
3-GIN-1-TREPH	115,758	115,758	115,791	115,758	115,758	115,758

**Table 5 entropy-25-00331-t005:** Graph classification results in the study of placing TREPH at different positions in GNNs. The best results within each group are emphasized in **bold** font.

Method	Pos	REDDIT-B	IMDB-B	IMDB-M	COX2	DHFR	NCI1
3-GCN-1-TREPH	1	94.0	77.0	**49.3**	78.7	80.0	76.6
	2	94.5	**79.0**	48.0	**83.0**	82.7	**78.4**
	3	94.5	77.0	**49.3**	78.7	82.7	77.6
	4	**96.0**	78.0	**49.3**	63.8	**86.7**	**78.4**
3-GAT-1-TREPH	1	**91.5**	**76.0**	**46.0**	**78.7**	**81.3**	**76.2**
	2	51.0	57.0	37.3	**78.7**	61.3	60.6
	3	50.0	50.0	33.3	**78.7**	38.7	50.1
	4	50.0	50.0	33.3	**78.7**	61.3	49.9
3-GIN-1-TREPH	1	90.0	77.0	47.3	80.9	81.3	78.8
	2	**94.5**	**79.0**	**50.0**	**89.4**	**88.0**	79.8
	3	91.0	78.0	48.7	83.0	**88.0**	79.1
	4	94.0	78.0	**50.0**	85.1	77.3	**80.8**

**Table 6 entropy-25-00331-t006:** Graph classification comparison results with state-of-the-art methods. The best results overall are emphasized in **bold** font. The best results in the group of TDA-based approaches are underlined. The results of all baseline methods except SAT are quoted from their papers, where “n/a” means “not available”. The experiments of SAT are implemented by us using the code of its paper, where “OOM” means “Out Of Memory”.

Method	REDDIT-B	IMDB-B	IMDB-M	COX2	DHFR	NCI1
*Kernel Methods*						
WL [72]	78.0±0.6	71.2±0.5	50.3±0.7	79.7±1.3	81.7±0.8	85.6±0.4
WL-OA [72]	87.6±0.3	74.0±0.7	50.0±0.5	81.1±0.9	82.4±1.0	86.0±0.2
*Neural Networks*						
PATCHY-SAN [73]	86.3±1.6	71.0±2.3	45.2±2.8	n/a	n/a	78.6±1.9
DGCNN [74]	n/a	70.0±0.9	47.8±0.9	n/a	n/a	74.4±0.5
SAT [75]	OOM	73.8±4.6	49.9±3.5	79.2±5.0	80.2±4.1	75.3±1.6
*TDA-based Approaches*						
SV [66]	87.8±0.3	74.2±0.9	49.9±0.3	78.4±0.4	78.8±0.7	71.3±0.4
PersLay [41]	n/a	72.6	52.2	81.6	80.9	74.0
GFL [42]	90.2±2.8	74.5±4.6	49.7±2.9	n/a	n/a	71.2±2.1
TOGL [43]	90.1±0.8	74.3±3.6	52.0±4.0	n/a	n/a	75.8±1.8
TREPH	$\underline{91.5\pm 2.2}$	$\underline{75.7\pm 3.6}$	$\underline{52.3\pm 1.6}$	$\underline{82.2\pm 3.0}$	$\underline{82.9\pm 3.0}$	$\underline{78.4\pm 1.5}$

## Data Availability

The data presented in this study are openly available in TUDatasets at https://chrsmrrs.github.io/datasets/ (accessed on 10 January 2023), reference number [71].

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
