# Peer review of "TREPH: A Plug-In Topological Layer for Graph Neural Networks"

_entropy, 2023, doi:10.3390/e25020331_

Round 1

Reviewer 1 Report

This is a well-formulated paper advocating and formulating the use of persistent homology in machine learning and neural networks. I recommend publication of the paper as it is.

Author Response

Response to Reviewer 1 Comments

Comment: This is a well-formulated paper advocating and formulating the use of persistent homology in machine learning and neural networks. I recommend publication of the paper as it is.

Response: We appreciate the effort of the reviewer.

Reviewer 2 Report

This manuscript proposed a new GNN layer which applies the   Extended Persistent Homology (EPH) methodology. The EPH layer extracts global geometry information (Topological Representation in the author's description) into the node feature. Experimental results showed better results when TREPH was adopted in GNN.

1. Computational complexity analysis including parameter quantity comparison is needed when inserting TREPH into a typical GNN.

2. The TREPH layer's input/output format and how to generate the topological representation are not described clearly enough. It is an important trick of applying TDA in GNN, readers may be interested in the details of TREPH so I suggest describing the TREPH in a better manner so that readers can fully understand your approach. Besides, it would be better if the source code can be available in github.

3. This paper compares their results with DGCNN presented in 2018. Many graph transformers with better performance than GNN appeared in 2022. It will be more convincing if compare with a graph transformer model since a graph transformer utilizes the global relationship of graph nodes with positional encoding which contains hidden geometry information.

Reviewer 3 Report

1.      Included in Algorithm1, there are places where you need to explain the physical meaning of each parameter.

2.      Do you think it is possible to add some explanations for the back-propagation during training?

3.      “First, the topological information extracted by PH is incomplete. Second, the infinite coordinates in the PDs turn out to be problematic for certain tasks.” These two questions which need to be solved at the beginning of this article do not echo in the experimental part.

4.      There is a problem with the graders on lines 207 and 209. You cannot use the symbol for first-level headings

5.      The meaning of each parameter symbol is best described as a table of parameter meanings, followed by abbreviations

6.      “Entropy” is one excellent journal, please study and cites some excellent related papers published on “Entropy”.

Round 2

Reviewer 2 Report

The revised manuscript added model parameter details and a comparison of results. Clarification of the algorithm is supplemented with a few modifications.

My questions have been answered and I have better understood the methods proposed in this manuscript. I think it is suitable for publishing now. The author's code is expected to be available.

Reviewer 3 Report

authors have addressed all my concerns

Author Response

Thank you!